# Shaping Policy on Chronic Diseases through National Policy Dialogs in CHRODIS PLUS

**DOI:** 10.3390/ijerph17197113

**Published:** 2020-09-28

**Authors:** Dorota Sienkiewicz, Alison Maassen, Iñaki Imaz-Iglesia, Elisa Poses-Ferrer, Helen McAvoy, Rita Horgan, Miguel Telo de Arriaga, Andrew Barnfield

**Affiliations:** 1EuroHealthNet, 1000 Brussels, Belgium; 2Instituto de Salud Carlos III–“Carlos III” Institute for Health, 28029 Madrid, Spain; imaz@isciii.es; 3Agency for Health Quality and Assessment of Catalonia (AQuAS), Government of Catalonia, 08005 Barcelona, Spain; eposes@gencat.cat; 4Institute of Public Health in Ireland, D08 NH90 Dublin, Ireland; helen.mcavoy@publichealth.ie; 5Directorate-General of Health of Portugal, 1049-005 Lisbon, Portugal; ritahorgan@dgs.min-saude.pt (R.H.); miguelarriaga@dgs.min-saude.pt (M.T.d.A.); 6School for Policy Studies, University of Bristol, Bristol BS8 1TZ, UK; andrew.barnfield@bristol.ac.uk

**Keywords:** policy, chronic diseases, deliberative dialog, complexity, health systems, public health, stakeholder engagement, policymaking, innovation, prevention, health promotion, health in all policies

## Abstract

Policy dialogs are deliberative dialogue that gather policy makers and relevant stakeholders from across disciplines to discuss a topic of mutual interest. They typically serve as a single element in a broader policymaking cycle, either informing the content of new policy or forming a component of policy evaluation and review. In the joint action CHRODIS PLUS, national policy dialogs were conducted in fourteen EU Member States. The aim of the dialogs was to identify new policies or changes to existing policies and legislation that are capable of tackling major risk factors for chronic disease, to strengthen health promotion and prevention programs and to ensure health systems are equipped to respond to priority issues within the chronic diseases field. In this paper, we present the CHRODIS PLUS policy dialog methodology, as well as results and lessons learnt from three national policy dialogs held in Ireland, Portugal and Spain. After discussion of the results, we conclude that the CHRODIS PLUS methodology is an effective mechanism to provoke deliberative discussion around chronic disease prevention and management in different countries. However, it is essential to ensure adequate human and financial resources—as well as political commitment—to accomplish objectives set out during the policy dialogs. We argue that priority-setting across sectors can improve the resilience of health systems and opportunities for investment in Health in All Policies (HiAP), both at European Union and Member State levels.

## 1. Introduction

### 1.1. The Challenge of Chronic Diseases in Europe and the Joint Action CHRODIS PLUS

National health systems require further resources and capacity if they are to meet the growing challenge of chronic diseases. Eighty-five percent of all-cause deaths in the European Union (EU) are due to chronic diseases [1]. In the EU in 2016, two-thirds of premature deaths of people under 75 were avoidable, meaning that 1.2 million of 1.7 million deaths could have been averted by reasonable means [2]. Of these, 741,000 deaths could have been prevented through effective public health and primary prevention interventions and 422,000 deaths could have been avoided through timely and effective healthcare interventions. While 80% of total healthcare costs are spent on managing chronic disease, with further associated expenses for social protection systems, an average of only 3% of national health budgets are allocated to preventative care/collective measures [3]. Reducing the incidence of chronic diseases is key to reduce the burden on health systems.

As a response to this challenge, the joint action (JA) CHRODIS PLUS was funded by the European Commission from 2017–2020 [4]. A joint action is a collaborative action among countries participating in the EU Health Program to “develop, share, refine, test tools, methods and approaches to specific issues and engage in capacity building in key areas of interest” [5]. The specific goal of CHRODIS PLUS is to support—using cross-national initiatives identified in the preceding CHRODIS joint action (2014–2017)—to reduce the burden of chronic disease while assuring health systems sustainability and responsiveness [6].

CHRODIS PLUS supports Member States in the implementation of new or innovative policies and practices for health promotion and disease prevention, management of chronic diseases and multimorbidity, as well as addressing social aspects such as employment and chronic diseases [4]. As part of this effort, national policy dialogs were held in fourteen EU Member States. The policy dialogs aimed to enhance the development of national policy responses on the prevention of chronic disease and served as a tool to strengthen the development of policies and strategies related to health promotion, disease prevention and innovative management of chronic diseases (Annex I). Policy developments were not prescribed from the outset but could include enhanced implementation of existing policy commitments or a change of focus or priority, as well as the emergence of new innovative policy options. Policy dialogs sought to provide a “safe space” for participants from different disciplinary or departmental backgrounds to discuss the challenges and opportunities for intersectoral cooperation in chronic disease prevention and management.

The aim of this paper is to explore the ways in which the CHRODIS PLUS policy dialogs were perceived and evaluated, planned and managed in the unique context of a European joint action and specifically in three national settings. It aims to establish what lessons can be drawn for the future use of policy dialogs as a deliberative dialog and upstream policymaking tool in health. Furthermore, this paper aims to contribute to the evolving literature on the utility of policy dialogs and their potential for enhancing country-level responses to chronic diseases.

### 1.2. Policy Dialogs and the CHRODIS PLUS National Policy Dialogs

A policy dialog is a type of deliberative dialog. Although policy dialogs play a crucial role in the policymaking process, they are often understood and applied in different ways and by different actors. The European Observatory on Health Systems and Policies, for example, defines a policy dialog as “an event where dialog takes place around “a policy question…on which…key documents and international experts… [are brought together] to present recent evidence, as well as relevant case studies from countries that have faced a similar question”” [7]. Policy dialogs are also described as “component [s] of the policy and decision-making process … intended to contribute to informing, developing or implementing a policy change following a round of evidence-based discussions, workshops, and consultations on a particular subject” [8]. Important elements inherent in policy dialogs include a certain amount of existing or potential alignment of “beliefs, values, interests and goals/strategies of elected officials and social interest groups” [9], the use of multidisciplinary teams to translate evidence into policy and a measure of transparency and accountability to ensure broad support [10]. Naturally, it cannot be presumed that a single policy dialog can change legislation or bring about a new strategy in a vacuum. Nevertheless, a well-organized dialog can contribute to such objectives. The policy dialog model has demonstrated to be particularly useful in times of transition to help study past achievements and explore strategic options for further health system reform [11].

The existing literature on policy dialogs has highlighted the value of studying the impact of collective decisions, actions, processes and goals on governmental and organizational decision-making and the use of evidence in policy [12,13,14]. Policy and governance literature [15] describes a policymaking cycle that includes: (1) defining problems [16] and agenda setting [17], (2) policy formulation, (3) decision-making, (4) implementation and (5) evaluation [18], which may or may not occur sequentially. Each stage is subject to complex interacting elements that can influence policy progression and similarly influence the effect of any given policy dialog [19]. Despite the growing literature that advocates for the value and implementation of deliberative dialogs such as policy dialogs, there has been insufficient evaluation of their influence on the policymaking process [20]. The prevention and management of chronic disease are complex policy issues and require cross-sectoral action addressing upstream as well as downstream determinants of disease [21]. Policy dialogs provide the opportunity to discuss not just the role of health policy, but also the potential for better coordination across government and stakeholders in line with a Health in All Policies (HiAP) approach [22].

The CHRODIS PLUS national policy dialogs were designed to either start policy processes or support ongoing policy developments. The overall expectation of the dialogs was to foster agreement on collaborative actions to respond to a defined policy issue and act as a catalyst for progress to implement those actions. The specific objectives varied across Member States and topics, including: (a) initiating or contributing to legislative change intended to help prevent or reduce the burden of chronic diseases, (b) establishing governance mechanisms for institutional/cross-sectoral collaboration/consultation, (c) reaching consensus on tangible actions that will address the identified problem and assigning responsible actors, (d) increasing political will and engagement towards new or adapted policies.

In CHRODIS PLUS, “deliberative” policy dialogs were used because they are a critical tool to aid the development of responsive, effective, sustainable and evidence-informed policy [23]. They gave participants a single medium through which they could address multiple influences on the policy process, such as the engagement of key stakeholders, the latest research evidence and existing policy and practice [19]. It is important to note, however, that feedback from participants in the CHRODIS PLUS national policy dialogs indicated that their efficacy as a tool would have been enhanced had the joint action envisioned providing sustained financial support to the dialog process, as well as a framework for monitoring implementation of related action plans over the short-, mid- and long-term. Current literature also supports the importance of sufficient financial commitment for policy dialog processes [24].

### 1.3. Outline of the Paper

This paper explores the practicalities of the CHRODIS PLUS national policy dialog planning and implementation processes and reviews the results of three dialogs in more detail.

The outline of this paper is as follows:In the Methods section, we set out the CHRODIS PLUS policy dialog methodology (see Appendix A for the participating Member States, dialog topics and national organizers).In the Results section, we present three case studies: the national policy dialogs organized in Ireland, Portugal and Spain. These dialogs were selected because they chose major modifiable risk factors for chronic disease as their primary themes, including tobacco, overweight and obesity and environmental conditions. They were also all working on the basis of existing policy or proposed legislation, which they wished to either make more effective or formally pass into law. The dialogs were held on: (1) reducing socioeconomic inequalities in smoking as part of the application of the Tobacco-Free Ireland policy and the Healthy Ireland Framework (Ireland), (2) reducing the growing rate of obesity and chronic disease with a particular focus on children and young people via a proposed law regulating food marketing and advertising (Portugal) and (3) alternatives for the effective implementation of health impact assessment as part of the application of the Spanish Public Health Act (Spain). In addition, these were among the dialogs with the highest response rates to the participant feedback questionnaire (see Section 2.6 and Table 1), giving us greater insights into perceived strengths and weaknesses of the respective dialogs.In the Discussion section, we build on the results and discuss the value of policy dialogs as a tool for enhancing policymaking, focusing on building coalitions, taking a HiAP approach, strengthening EU added value of national policymaking and standardizing policymaking processes.Finally, we present our conclusions from the national policy dialogs and suggestions for further research.

## 2. Methods

### 2.1. Organizing a Policy Dialog

The key features of a deliberative dialog are: (1) the meeting environment (enabling or inhibiting), (2) the mix of participants (expertise, experience, interests, capacity) and (3) the role of evidence [25]. Deliberative approaches like those used in policy dialogs foster structured conversations that promote listening as much as speaking, in order to feed into an informed and reasoned agreement. Participants are encouraged to explore the values that may underlie opposing views and weigh reasons for and against different policy options, with the aim to determine mutually agreed action [26]. These approaches enable participants to explore diverse views, build relationships of trust and improve their understanding of the working context of other key stakeholders. They also connect research to personal/lived experience, including patients’ voices and help participants to develop agency to take action within their own spheres [27].

The CHRODIS PLUS guide for the national policy dialogs was developed to incorporate key features of deliberative dialogs and to explain essential enabling requirements of the dialogs [28]. These key features include defining roles and responsibilities, preparing responses to the CHRODIS PLUS questionnaire (part of the CHRODIS PLUS guide for national policy dialogs), preparing the meeting agenda and inviting participants, reporting the dialog and preparing the action plan and evaluating the dialog’s implementation (through the use of participant feedback questionnaires). Each step is further described below.

If the National Policy Dialogs needed ethical approval, the national legislation was strictly followed. The subjects gave their informed consent for inclusion before they participated in the National Policy Dialogs. If the ethical approval was needed, the National Policy Dialogs were conducted in accordance with the Declaration of Helsinki and the protocol was approved by the respective Ethics Committee. Grant Agreement: 761307—CHRODIS-PLUS: Implementing good practices for chronic diseases (CHRODIS-PLUS). The joint action CHRODIS-PLUS is funded by the Health Program 2014–2020 of the European Union.

### 2.2. Assignment of Predefined Roles and Responsibilities

The three key roles in the CHRODIS PLUS national policy dialogs were:aNational organizer

The national organizer had the primary responsibility for the overall design and execution of the dialog, including selection of the topic, invitation of participants and compiling the final report and action plan. They also played an important role in managing the expectations of the participants by communicating the aims of the dialog, providing background material and questions in advance of the discussion and clarifying the rules of the dialog.

bModerator

It was the responsibility of the national organizer to arrange for professional (external or internal) moderation. The role of the moderator was to trigger, maintain and conclude the discussion while directing the participants to agree on tangible actions. Ideally, the moderator was an external person to the national organizer with both thorough knowledge of the dialog topic and experience in moderation and facilitation. Knowledge of the topic allowed the moderator to direct the conversation, as needed, towards a more in-depth examination of barriers, facilitators and potential avenues for addressing the identified problem.

cRapporteur

The role of the rapporteur was to accurately record and report the discussion of the policy dialog. The rapporteur was tasked with clearly identifying the tangible actions that will address the identified problem. The use of Chatham House Rules, which allow for disclosure of information without identifying the source of the information (e.g., identity and affiliation of the speaker(s) and other participants), helped establish an environment that allowed a full and frank discussion of the identified problem [29].

### 2.3. Questionnaire Planning

A purpose-built questionnaire (Appendix A) was completed by each national organizer. There were several essential components for participants to consider:Clearly defining the objectives of the dialog. This goes hand-in-hand with a clear vision of what outcomes and results would be expected.Conducting a comprehensive context analysis as part of a problem definition to select main points for discussion.Conducting an effective stakeholder analysis (as a part of collecting evidence-based background information) to select policy dialog participants. As further described below, participation was invitation-only and limited to build trust and ensure more frank and open flow of conversation.Identifying a moderator who would be able to provide effective moderation and facilitation. This was key to having meaningful and comprehensive discussions.Defining tangible and feasible actions or steps to achieve the expected results of the dialog. This was done through an action plan (further detailed below). Actions may or may not occur during the period of the joint action.Ensuring follow-up and the will to implement. The policy dialogs were not intended to be “one-off” activities, but rather the initiation or continuation of a broader policymaking process.Describing the potential added value of their actions to wider European efforts to prevent and/or manage chronic disease.

The national policy dialogs were conceptualized and planned on the basis of the questionnaire. The topic, objectives and desired outcomes for each national dialog were reviewed and refined in iterative and close collaboration with the CHRODIS PLUS task leaders. Each topic was context-specific to the Member State, but there was some overlap across different countries.

### 2.4. Meeting Agenda and Participation

It is important to highlight that the national policy dialogs were not intended as dissemination events. The main goal was to engage a small group of “influencers” and senior change-agents in a practical and solution-oriented policy discussion. The numbers of non-essential attendees were limited in order to encourage exchange and trust and keep the discussions focused. Chatham House rules were followed in all dialogs to encourage frank and open discussion of complex and sometimes contentious issues.

Following the stakeholder and context analyses, participants were selected and invited from sectors relevant to the topic and desired outcomes. Priority was placed on attracting relevant high-ranking officials to bring more “political capital” to the dialog and official invitation letters were signed by the CHRODIS PLUS Coordinator. An appropriate venue was selected to maximize attendance and the policy dialog was conducted in the national language. Exceptionally, in particular when it concerned an external–yet essential–participant setting out the EU background/relevance, an introductory session was conducted in English.

Policy dialog documents were sent to the confirmed participants at least two weeks prior to the dialog and included a detailed draft agenda and pre-prepared question list, as well as further background materials, as appropriate. The pre-prepared questions aimed to: (1) introduce the identified issue (problem definition), (2) link the broad themes to more specific topics (an overview of current policy and program responses), (3) and address the key/core issues which would be explored in the policy dialog in order to elicit tangible and workable actions by the end of the meeting. This meant the discussion could start from a place of common understanding of the problem (e.g., inequalities in tobacco use) and move quickly into debate on potential response options.

### 2.5. Reporting and Action Plan

The policy dialogs were intended to be a single element in a broader policymaking cycle, either informing the content of new policy or forming a component of policy evaluation and review. For this reason, it was essential that each national organizer prepare for each dialog in advance by providing appropriate contextual information and resources, as well as continue after the dialog by preparing a report with the dialog minutes and an action plan outlining tangible and feasible next steps to address the identified problem. For each step of the action plan, dialog participants were assigned roles and suggested timelines were proposed.

A pre-structured reporting template was provided to the national organizers in advance of the dialog. This template complemented the national organizers and moderators in their preparations for the dialog. All reporting templates were submitted to the project leaders within one month following the completion of the dialog. Reports and proposed action items were then reviewed for clarity and feasibility by CHRODIS PLUS task leaders.

### 2.6. Evaluation

A common participant feedback questionnaire was designed to be sent to participants at the end of each policy dialog (Appendix A). The aim of the questionnaire was to assess the satisfaction of participants with the organization and outputs of the meeting, as well as compiling personal opinions about the main national barriers and facilitators to adopt the measures and recommendations discussed during the policy dialog. The CHRODIS PLUS evaluation leader analyzed the questionnaire results and shared a compiled report with project leaders and national organizers. This report had two main objectives: (1) to receive feedback about the meeting and (2) to identify aspects that could be improved in future policy dialogs in order to generate the best discussions and outputs. Finally, the survey was sent to participants in ten of fourteen policy dialogs. The survey response rates are included in Table 1.

## 3. Results

While fourteen national policy dialogs were performed (Appendix A), three were selected as case studies for this article on the basis of their selected topics and the high response rate to the participant feedback questionnaire (as described above).

### 3.1. Ireland

The Irish policy dialog was held on 12 June 2018 at Europe House in Dublin. The dialog topic was “Socioeconomic Inequalities in Tobacco Use”. Annually, 5900 people die in Ireland of causes related to tobacco use (about one in five of all deaths and the largest single cause of preventable death) [30]. Tobacco use drives an excess of acute and chronic ill-health and disability as well as premature death, particularly in disadvantaged communities [31]. In 2013, the economic cost of tobacco consumption in Ireland was estimated at €1.5 billion [32]. Tobacco use contributes to the generation, maintenance and deepening of poverty and diminished social and economic opportunity through both direct and indirect mechanisms [33].

The objectives of the dialog included the following: (1) identifying and exploring which elements of the Tobacco-Free Ireland program (established in 2013) were currently targeted to address socioeconomic inequalities in tobacco use, (2) exploring how European partnerships and initiatives could be leveraged in the future to support the reduction of inequalities in tobacco use in Ireland and (3) identifying which policy and program actions should be sustained and what new actions should be considered to address inequalities in tobacco use in the future.

The dialog brought together participants from the Tobacco and Alcohol Control Unit—Department of Health; Health Service Executive; Irish Cancer Society; Institute of Public Health in Ireland; and the Department of Public Expenditure and Reform (the department responsible for the development of efficient public spending, including equality budgeting). The following conclusions were reached by dialog participants: (1) approaches to addressing inequalities in tobacco use must be effective in targeting and addressing both prevention and smoking cessation; (2) tobacco pricing is central to addressing inequalities in smoking—but not enough on its own; (3) tobacco and social disadvantage is a cross-government agenda and integration in the operation of statutory services may be beneficial; (4) changing attitudes and norms around tobacco in disadvantaged communities is important, but can be challenging; (5) knowledge on what works for disadvantaged groups is evolving, but incomplete—this creates difficulties in decisions on investment of resources; (6) partnerships are critical to success in the health inequalities component of tobacco control policies and programs; (7) monitoring and accountability on health inequality dimensions is important; and lastly that (8) effective advocacy is critical to make progress on inequalities—but some of those most vulnerable to tobacco related harm are underrepresented.

Based on the conclusions reached, the Irish policy dialog action plan detailed policy options for the following four action points for further collaboration:(1)*Build tobacco into government considerations for equality budgeting.* The Department of Public Expenditure and Reform agreed to link with the Department of Health to consider the equity impact of fiscal policy on tobacco. Equality budgeting involves providing greater information on the likely impact of budgetary measures across a range of areas such as income, health and education and how outcomes differ across gender, age, ethnicity, etc. Equality budgeting helps policymakers to better anticipate potential impacts in the budgetary process, thereby enhancing the government’s decision-making framework. Coupled with Ireland’s commitment to keeping tobacco control policy free of any influence from the tobacco industry (in line with the WHO Framework Convention for Tobacco Control), equality budgeting can help highlight the societal returns from investing in equity-focused tobacco control legislation and policy.(2)*Build a more defined health inequality dimension into the development of clinical smoking cessation guidelines based on best evidence*. At the time of the policy dialog, the Health Service Executive was leading a process to develop updated Clinical Practice Guidelines on the management of tobacco addiction. This process included an objective and comprehensive review of international evidence as well as a consideration of the national context for service delivery. Subsequent to the policy dialog, a presentation on evidence relating to health inequalities and the conclusions of the policy dialog were presented directly to the clinical practice guideline development group.(3)*Enhance efforts to target investment in tobacco control including resourcing for equity-focused smoking cessation and progressive tobacco taxation*. All partners agreed on the importance of seeking opportunities to access additional resources to advance equity-focused smoking cessation, as there are ongoing challenges with securing sufficient resources in this field. Participants acknowledged that health promotion and public health/prevention budgets are constrained and there are competing priorities and that advocacy groups are critical to influencing at political level to ensure that tobacco control is understood and prioritized. Since the policy dialog, the Health Service Executive has enhanced their work on health equity in smoking cessation, while tobacco taxation in Ireland has continued to rise.(4)*Greater engagement with disadvantaged groups, in particular with people with mental health difficulties.* Participants agreed that there is a need to better understand the lived experience of socially disadvantaged people in their journey in and of tobacco addiction. Co-design with service users can be critical to engaging disenfranchised people and communities with state run smoking cessation supports. The findings of a comprehensive research project on this issue were discussed by the Clinical Practice Guidelines Development Group on smoking cessation. An engagement exercise (a conversation café event) with service users was also developed and held with mental health service users following the policy dialog [34].

The evaluation of the dialog found that the background information shared with participants, organization and facilitation of the dialog were rated as “excellent” by 75% of respondents. Other positive aspects participants cited were the inclusion of a member of the Department of Public Expenditure and Reform and the diversification of smoking cessation strategies in order to reach vulnerable groups. The perceived main barriers for the national implementation of the outcomes/proposals agreed in the policy dialogs were a lack of resources (specialist, supporting structures) and budget/funding issues. Further conclusions from the Irish policy dialog can be found in Table 2.

### 3.2. Portugal

The Portuguese policy dialog was held on 30 January 2019 at the offices of the Directorate-General of Health in Lisbon. The topic was “Advertisement of Food and Beverages to Children”. In Portugal, 86% of the burden of disease is due to chronic diseases and 15.8% of the years of healthy life lost are due to inappropriate eating habits. Thus, 41% of the total years of healthy life lost due to premature death could be avoided if we eliminated the major modifiable risk factors, such as overweight, sedentary lifestyle and poor eating habits, tobacco and alcohol use [35]. Portugal has been tackling childhood obesity in many different ways over the past years. The WHO COSI (Childhood Obesity Surveillance Initiative) study has shown a decrease in childhood overweight and obesity by almost 8% between 2008 and 2019, but more can be done. As of 2019, one-third of Portuguese children remained overweight or obese, a problem associated with inadequate eating habits: 69% of children do not consume the WHO-recommended 400 g of fruit and vegetables and 43% consume more than one soft drink on a daily basis [36].

Food advertising to children promotes products that are typically high in sugars and fat. The evidence indicates that unhealthy food and beverage marketing increases dietary intake and preference for energy-dense, low-nutrition foods and beverages [37]. The data supports public health policy action that seeks to reduce children’s exposure to unhealthy food advertising [38]. Food literacy, beginning early in life, is key to lowering the percentage of overweight people in Portugal and, consequently, decreasing the prevalence of chronic diseases among the adult population. Several WHO strategic documents recommend that Member States implement measures to reduce the impact of non-food and beverage marketing and advertising, namely the WHO European Food and Nutrition Action Plan 2015–2020 [39], the Vienna Declaration on Nutrition and Noncommunicable Diseases in the Context of Health 2020 [40] and the Report of the WHO Commission on Ending Childhood Obesity [41].

Against this backdrop, the main objective of the Portuguese policy dialog was to gather a group of national stakeholders together in order to find ways to tackle the issue of advertisement of food and beverages to children at national level. The specific objectives included: (1) understanding the perception of each participating entity regarding the current status of the matter, (2) understanding what each participating entity has done to date to address the matter, (3) discussing the effectiveness of regulation versus self-regulation, (4) discussing main barriers to regulation of food and beverage advertisements to children, and finally (5) understanding how each participating entity could contribute to next steps to tackle the issue. Participants in the dialog came from: the Directorate-General for Health; the Directorate-General for Consumer Affairs; the People–Animals–Nature Political Party; the Ministry of Agriculture; the Portuguese Regulatory Authority for the Media; the Directorate-General for Education; the Portuguese Association for Consumer Protection; the Food Safety and Economic Authority; and the Portuguese Institute for the Ocean and Atmosphere.

The following conclusions were drawn by the end of the dialogue: (1) all entities present in this policy dialog were directly or indirectly working on issues related to this subject; (2) in general, all entities agreed with self-regulation, but considered that it should coexist with regulation; (3) the role of health literacy is important in changing behavior among children, young people and families; (4) digital media creates new challenges and barriers for the regulation of advertised content; (5) the measures to be applied must focus not only on the media and digital platforms, but also on activities/actions that attract children and young people, such as festivals, merchandising and sporting events, among others; (6) the age of 18 years was selected as a cutoff for the target population (though eventually the cutoff was changed to 16 years old); (7) it was agreed that digital influencers should be used as partners for promoting healthier food choices; and that (8) “counter advertising” methods were also seen as very important to demystify traditional advertising practices.

The action plan detailed two action points, both oriented towards the creation of working groups:(1)*A working group focused on finalizing a legislative proposal regarding advertisement of food and beverages to children.* This proposed legislation aimed to limit advertising of foods and beverages with a high content of sugar, fat or sodium in preschool, basic and secondary education establishments, in children’s playgrounds and within a specific radius of those places, as well as in publications, programs or activities for minors. It also aimed to prohibit advertisement of such foods in the time before and after television programs which have a certain percentage of the audience below 18 years of age. The members of this working group met regularly to monitor and advocate for progress towards completion of this regulation. The legislation was successfully passed into law on 23 April 2019 with additional regulations approved on 21 August 2019 [42]. This working group has therefore completed its tasks.(2)*A working group focused on the promotion of health and food literacy among the Portuguese population (the “National Health Literacy Commission”)*. The objectives of the group were to define health and food literacy actions with two aims: (1) to improve the health and food literacy of the population and (2) to work together with digital influencers in order to raise awareness about health and food literacy. The working group was created, but its tasks have been put on hold due to the COVID-19 pandemic. Over time—and with the help and contribution of the National Program for the Promotion of Healthy Eating—further focus will be made on promoting healthy eating habits among the Portuguese population. For example, the Directorate-General for Health of Portugal (DGS) is currently undergoing a study among Portuguese families to verify in which level the legislation is really being respected by the industry and by all the different communication channels. DGS will implement a pilot study based on “CLICK: The WHO Europe framework to monitor the digital marketing of unhealthy foods to children and adolescents” [43].

The participant feedback questionnaire results found that the organization and the technical conditions of the meeting were excellent, receiving an average rating of 9.1 of 10. The participant stakeholders very positively expressed their will for continuing working on the topic together and in other scenarios/events. Further conclusions from the Portuguese policy dialog can be found in Table 2.

### 3.3. Spain

The Spanish policy dialog was held 10 June 2019 at the offices of the Ministry of Health, Consumer Affairs and Social Welfare in Madrid. The topic was “ALTERNATIV35: Health Impact Assessment. Alternatives for an effective implementation of Article 35 of the Spanish Public Health Act 33/2011” [44]. The topic was selected because of the need to further develop the Article 35 of the Spanish Public Health Act, which obliges Spanish Public Administrations to perform Health Impact Assessments (HIA). The Dialog took place between the Ministry of Health, Consumer Affairs and Social Welfare (MoH) and the Ministry for Ecological Transition (MET).

The main objectives were to reinforce collaboration between the MoH and the MET to jointly address health and environment issues and to assess the possibility to include health impact assessment in environmental projects and measures. They specifically hoped to achieve a clear commitment to intersectoral work addressing equity and social and environmental determinants of health, as well as the establishment of formal mechanisms for joint work between the two ministries.

The dialog drew participants from: the Directorate of Public Health, Quality and Innovation (MoH); the Directorate of Biodiversity and Environmental Quality (MET); the Carlos III Health Institute; the Ministry of Science and Innovation; the Sub-directorate of Health Promotion and Public Health Surveillance (MoH); the Sub-Directorate of Environmental Health and Occupational Health (MoH); the General Secretary of Health and Consumer Affairs (MoH); the Sub-Directorate of Environmental Assessment (MET); the Sub-Directorate of Air Quality and Industrial Environment (MET); and the Sub-Directorate of Quality and Innovation in Health (MoH).

The discussion evolved around three core points: (1) exploring the views of the participants toward the links between health and environment and possible synergies, (2) improving the integration of health in environmental policies and in environmental impact assessments and (3) how participants rated the HIA recommendations provided in the background document and preparatory work conducted prior to this dialog.

Participants positively received the participatory work, and the following priority measures for the integration of health into environmental policies were agreed: (1) participation of technical staff from the health departments in consultations on Environmental Impact Assessments (EIA); (2) defining the areas in which to carry out the HIA; (3) establishment of indicators and criteria for HIA; (4) technical advances in the establishment of relevant measurements and thresholds for health impact control; (5) overcoming the limitations for the mobility of technical staff among the Ministries involved; (6) participation of technical staff from the health departments in the preparatory reports prior to legislative proposals; (7) establishment of methodological guidelines to perform EIA; (8) reasonable criteria that prevent public administration to carry their day-to-day activities because of insufficient evidence; and (9) establishment of procedures to evaluate plans and programs that are not currently evaluated through EIA.

The action plan detailed just one action point: an agreement to create an interdepartmental working group between the two Ministries to address the above-mentioned priority measures. This is a permanent working group with participants from both Ministries (Ministry of Health, Consumer Affairs and Social Welfare and Ministry for Ecological Transition).

The evaluation of the dialog found that participants highlighted the importance of the topic and valued very positively the communications, interaction and collaboration between the Ministry of Health, Consumer Affairs and Social Welfare and the Ministry for the Ecological Transition. Further conclusions from the Spanish policy dialog can be found in Table 2.

## 4. Discussion

On the basis of the results of the national policy dialogues—and in particular the three selected case studies—several important themes emerged. These relate to affirming the value of policy dialogues as a deliberative dialogue approach for health policymaking. Specific uses of the policy dialogue as a tool for enhancing policymaking are also discussed. These include the value of using policy dialogues as a tool to bring together key stakeholders, to support a Health in All Policies (HiAP) approach and to improve the transparency of policymaking processes. In addition, we found that the policy dialogs were a particularly relevant activity in the context of a joint action, which both aims to build individual Member State capacities in the selected area (e.g., prevention and management of chronic disease), while also demonstrating and enhancing EU added value from the collaboration. Finally, common strengths and weaknesses of the particular CHRODIS PLUS methodology—in the context of JA CHRODIS PLUS—are discussed.

### 4.1. Policy Dialogs as a Tool to Support Coalition Building, Address Complexity and Adopt a Health in All Policies Approach

Chronic diseases are the result of a combination of genetic, physiological, environmental and behavioral factors [45]. The complexity of chronic disease prevention and management presents a “wicked” problem, one that can only be resolved by navigating potential conflicts of goals/interests/stakes and important technical disputes, as well as engaging multiple actors from several levels of government [12]. Resolving such problems requires policy leadership and commitment and cross-sectoral engagement across such domains as health, fiscal, social, employment and the environment. Broader coalitions of stakeholders reduce the fragmentation and “silo” mentality that has often hampered intersectoral solutions [15]. Yet, there is a broad lack of good practices in evidence-informed policymaking for complex challenges such as prevention and treatment of chronic diseases [46,47].

In the past decade, it has become clearer that policymaking processes should espouse more deliberative approaches, reconfiguring the traditional models of policy development built around interests, institutions and existing policies [19]. The literature on coalitions points towards the need for careful discussion, deliberation and debate around goals, targets and issues, an open and inclusive process, a recognition of varying levels of contribution and capacity and transparent reporting and recording [20,21]. Tools that seek to think with complexity promote a better understanding of the wider political, institutional and cultural contexts in which health outcomes, risk factors and behaviors are embedded [19,48]. The findings of our research with policy dialogs for health promotion supports the work of Ayana and colleagues, whereby, the policy dialogs are a space in which non-state actors can document and effectively communicate field evidence, form strong networks with state actors and establish alliances with key decision-makers [13].

As Table 2 demonstrates, a key strength across the three case studies was the ability of the policy dialogs to facilitate interactions between different sectors—notably between health, finance and environment. The dialogs set the groundwork for future joint activities (e.g., the action plan, working groups). Yet, participant feedback from each case study also indicated that there was a perceived need to engage still more stakeholders, particularly from the target populations who would primarily be affected by the proposed policies. Facilitating more interactions between policy makers, influencers and target populations can help identify the intervention points and the relative contribution of different stakeholders and pursuing a systems approach encourages action at the most crucial points in the system [46]. Adopting a complex systems approach means navigating away from simple and linear causal models towards thinking how processes and outcomes within a system facilitate change [21].

HiAP and addressing complexity require a new suite of tools compared to those traditionally used in the development of policy. They also entail more focus on implementation and solid evaluation processes at short-, mid- and long-term [45]. We have found that the use of policy dialogs can encourage new alliances, connections and deepen working relationships between sectors. We have found that policy dialogs can provide a new way to empower thinking away from isolated intervention approaches and reconfigure governance by breaking departmental silos and facilitating cooperation and a sense of shared purpose with broader groups of stakeholders. Our findings complement research conducted by Ongolo-Zog et al. as dialog participants engaged with new information and moved to decision points following the dialogs [14] and new policy issue networks emerged in Ireland, Portugal and Spain as a result of the dialogs.

### 4.2. Policy Dialogs as a Tool for Enabling National Policy Development through European Cross-Country Collaboration

Rising levels of chronic disease are a significant challenge for all European Member States, but Member States vary considerably in their policy responses [49]. These different responses were reflected in the wide range of topics and diverse types of stakeholders involved in the CHRODIS PLUS national policy dialogs. At the same time, however, preparing for the national policy dialogs provided national organizers with an opportunity to exchange and learn from other countries as well as reflect on the European common frameworks and any precedents from across Europe that may support their efforts. In the case of the three selected case studies, each built on existing policy efforts inspired by broader European or EU Member State policy developments. For instance, Tobacco-Free Ireland built on the WHO Framework Convention for Tobacco Control and a number of EU Directives and Recommendations [50]. National initiatives to regulate advertising to children had been undertaken in countries such as Denmark, France, Spain and the United Kingdom before the Portuguese regulation was drafted [51]. In preparations for the Spanish policy dialog, a review of proposals for health equity assessment, including from Spanish autonomous communities as well as the joint action “Equity Action” was conducted [52].

Thus, national policy dialog topics were often influenced by wider trends and policies happening in the EU. The overlap seen in selected topics across policy dialogs (Appendix A) suggests that CHRODIS PLUS activities could directly help Member States to address similar concerns and to learn from one another’s experiences to contribute to more successful implementation of new policies or policy revisions. National organizers in the three case study countries found that this connection with the wider “European” identity—as part of an EU co-funded joint action—brought more value and credibility to the policy dialog process. Having project leaders from CHRODIS PLUS participate in each dialog, and, in the case of Ireland, hosting the dialog in Europe House, also underscored the potential contributions that each policy dialog’s outcomes could have in influencing other decision-makers across Europe. The dialogs therefore proceeded in line with the specific goal CHRODIS PLUS to support Member States to use cross-national initiatives to reduce the burden of chronic disease while assuring health system sustainability and responsiveness [6]. The CHRODIS PLUS policy dialogs offered a mechanism for knowledge translation and exchange within and between EU Member States, particularly around decision-making in health systems [19].

### 4.3. Policy Dialogs as a Method for Enhancing Transparency and Standardization of Policymaking Processes

We argue that policy dialogs can contribute to bridging the gap between policy and operational delivery. The CHRODIS PLUS policy dialog methodology relies on thorough stakeholder and context analyses prior to the dialog, allowing participants to review the latest evidence and come to table with a common understanding of the topic at-hand. This can facilitate more evidence-based and transparent decision-making procedures, with the potential to reduce the “black box” of traditional policymaking [23]. Our experience of using the CHRODIS PLUS policy dialog format shows that it could be adapted for use in different contexts and at different stages of the policymaking process. It could be used not only to examine and refine existing policies, but also to “kickstart” discussions around emerging areas of policymaking. They could also be used not only to translate high-level policy goals into national legislation, but also to initiate new programs, services and community engagement. Our analysis of the participant feedback and action plans from the national policy dialogs, including the three case studies, identified the following “success factors” for CHRODIS PLUS national policy dialogs:Careful and “fit-for-purpose” stakeholder engagement prior to, during and after the dialog event;Close liaison between national dialog leads and the CHRODIS PLUS task leaders;Preparation and planning on supporting documentation including questionnaires, briefing documents, reporting and evaluation;Careful structuring and facilitation of dialog discussions, including the follow of the Chatham House Rules;Agreement and common writing of conclusions by participants;

The engagement of key stakeholders seemed to have a crucial role in the planning and implementation of new approaches [20]. Policy dialogues were useful in the context of implementation science and in bridging the gap between policy and strategy level goals and actions and the “real world” challenges of operationalizing those goals and actions. Stakeholders were not only limited to those directly involved during planning, but also to those that have a primary and secondary role when implementing the practice, particularly for ensuring its sustainability over time. Finance, marketing or environmental departments are examples of stakeholders whose participation in the policy dialogs was highly valued by attendees. These stakeholders seemed to create further awareness of potential barriers to implementation related to access to resources, reaching vulnerable groups, ethical concerns and potential competitors to new approaches, sometimes not fully known by the developers and implementers of new health strategies. The policy dialog also highlighted where evidence to guide investment in policy and practice was lacking or where there were gaps in knowledge translation in the research community. The creation of new synergies between those stakeholders, as well as establishing stable communications and coordination (e.g., working groups) between them seemed to be one of the most accepted approaches to ensure successful design of policies oriented to contextual needs and sustainable implementation.

Further analysis of the participant feedback and action plans also revealed the most relevant common difficulties highlighted in the policy dialog process (some of these points have been illustrated in Table 2). Barriers that inhibit the implementation of new practices include: a lack of supportive legal frameworks and measures, weak or old regulations, poor diversification of strategies and ambiguity on technical, human, legal and financial resources needed for implementation. Three key approaches were highlighted during the policy dialogs in order to create greater impact and sustainability on the design and implementation of new practices: (1) knowledge that self-awareness of target population was not enough and therefore marketing, communication and diversity of approaches were needed in order to reach vulnerable groups; (2) necessity of creating stable working groups, “policy partnerships” and relationships with people from across different departments and policy fields, lobbies and associations involved in the implementation/operationalization for a continuous sharing of communication and information; and (3) creation of evaluation strategies of the implementation in order to share transparent results to policy makers and stakeholders, making timely adjustments when necessary.

## 5. Conclusions

This section summarizes key learnings from the preparation, organization and implementation of the CHRODIS PLUS national policy dialogs. We also suggest some recommendations for future action.

The CHRODIS PLUS policy dialog methodology proved an effective mechanism to provoke deliberative discussion on a wide range of policy topics in different settings.

The dialogs generated added value by stimulating national thinking and concrete actions, including post-facto, about priorities and rationales to address chronic diseases. The questionnaire, reporting, preparation of action plans and evaluation of the feedback from participants helped keep stakeholders engaged, raise their awareness of needs, challenges and opportunities, as well as set concrete goals and objectives for a wide variety of subjects in a context of chronic diseases.

The long-term uptake and eventual scale-up of good practices in health promotion and disease prevention can hardly be achieved without a multilevel and multisectoral approach. Our research adds further support to usefulness of an inter-sectoral approach to health promotion and disease prevention as key to addressing chronic diseases. Given that many of the determinants of chronic disease lay outside of the health sector (e.g., environment, education, socioeconomic status), all dialogs acknowledged the importance of bringing together and engaging stakeholders from outside of the health sector. We have managed to effectively use a deliberative debate tool such as policy dialogs to increase stakeholders’ engagement. Most action plans laid out specific activities or areas of further work that would bring in other stakeholders in future actions.

Using a framework like the CHRODIS PLUS policy dialog methodology can support future policy makers in such activities as identifying key stakeholders, preparing background documentation, selecting the optimal points in the policymaking process for deliberative dialog and implementing and evaluating policy dialogs.

Ensuring adequate human and financial resources to accomplish objectives set out during the policy dialogs– and working to gain and maintain political commitment—are essential.

An important structural challenge to the CHRODIS PLUS policy dialog process related to a lack of resources allocated for follow-up activities and insufficient monitoring (e.g., mid- and long-term evaluation) of the implementation of national action plans. This lack of follow-up meetings and processes falling outside of the scope/timeframe of the joint action makes it challenging to ensure continuity of any processes initiated during the policy dialogs. However, two of the three selected case studies have already managed to continue work beyond the fixed temporary framework of their action plans. We conclude that the sustainability can be improved if all conceived actions are integrated into existing policies, programs or processes. In addition, political capital and commitment—and particular, sustained funding—were seen as key to achieving many of the follow-up actions indicated by the policy dialogs [24]. We suggest that further research of this aspect could result in important longer-term learnings about policy dialog planning, implementation and overall impact. Understanding how priorities and political landscapes adapt over time, it is also important that dialog participants envision ways to gain and maintain political commitments from the necessary policy- and decision-makers.

Future projects and programs, including joint actions, which rely on policy dialogs could use these learnings to inform the design and funding of their policy dialog processes. Overall, more research on effective policy dialog design, informed by the latest literature on deliberative dialogs, could help standardize policymaking procedures and improve network governance through increased dialog and civic engagement [20,22,27].

Priority-setting and implementation across sectors can improve the resilience of health systems and opportunities for investment in Health in All Policies, both at EU and Member State levels.

Given that many of the determinants of chronic disease lay outside the health sector (e.g., environment, education, socioeconomic status), an effective policy response to prevent and manage chronic diseases must improve communication and coordination across all sectors and all levels. Working together to establish a common understanding of challenges and to jointly set mutually beneficial priorities can reduce the burden on the health system, enhance efficiency of social service provision and increase opportunities to fund new and innovative approaches [53]. While investments in health and wellbeing are first and foremost a government responsibility, many European programs also invest in health (including indirectly through other social sectors). Unfortunately, this information is dispersed across many ministries (e.g., health, education, finance, justice, environment). The CHRODIS PLUS policy dialogs showed a need for facilitated inter-ministerial and vertical communication on opportunities for improving health policy and practice within the EU and its Member States.

Improving awareness of actions which can be taken at national level (e.g., joint health and environmental assessments, equality budgeting techniques) can increase the impact and perceived ownership of health policies among key stakeholders. Improving awareness of EU funding opportunities, such as joint actions, can also enhance the value of these programs across all levels of governance. We recommend strengthening the EU’s national focal points (for health) and improving their cooperation across sectors (e.g., bundling of projects). We also suggest that the European Commission’s Steering Group on Health Promotion, Disease Prevention and Management of Noncommunicable Diseases could be better leveraged to facilitate inter-ministerial and vertical communication to enhance HiAP approaches at national and European levels [54].

## Figures and Tables

**Table 1 ijerph-17-07113-t001:** Summary of the feedback questionnaire respondents from ten of fourteen national policy dialogs of CHRODIS PLUS.

Country	Participants	Respondents	Response Rate
Croatia	15	7	47%
Hungary	12	4	33%
Ireland	7	4	57%
Italy	17	7	41%
Malta	10	8	80%
Poland	19	8	42%
Portugal	17	11	65%
Slovakia	11	3	27%
Slovenia	16	7	44%
Spain	18	11	61%

**Table 2 ijerph-17-07113-t002:** Summary of the main conclusions of the national policy dialogs in Ireland, Portugal and Spain.

Country	Ireland	Portugal	Spain
Title of the PD	Tobacco control and inequalities	Advertising of foods directed at children	Health Impact Assessment (HIA)
Most relevant topic discussed	Diversification of smoking cessation strategies in order to reach vulnerable groups. Evidence for best practice in reducing inequalities and testing of bespoke interventions Continued attention to accessibility and affordability of tobacco including taxation. Integrating tobacco control as a poverty prevention and response measure across government. Enhanced smoking cessation and listening engagement with vulnerable groups including people with mental health issues.	Need for a formal legal framework on publicity for children, regulating the (digital) marketing of foods and beverages. Awareness that self-regulation is not enough; Need for literacy improvement among the population.	Reinforcement and collaboration between health and environment sectors in health impact assessments in environmental projects and measures. Health and equity in all policies. The firm commitment of the Ministry of Health and the Ministry of Environment to work together.
Main barriers	Lack of resources: specialist, supporting structures, etc. Budget/funding issues. Difficult to reach vulnerable groups. Competing pressures. UK leaving the EU via UK-influence on practice, policy and research, resources and partnerships.	Power of the food industry. Existence of lobbies. Delay of the legislative process. New challenges of digital media as attract children and young people through different activities.	Lack of technical, human, legal and financial resources for implementation. No specific legal measures yet developed to support actions. Decentralization. Awareness among citizens about contribution to their own health and importance of changing habits.
Main facilitators	Political support. Evaluation milestones within policy delivery. Involvement of multiple government departments and sectors. Collective leadership and partnership through the Tobacco-Free Ireland partners group. National and European partnerships.	Common understanding of the situation, needs and problem to solve among different stakeholders, including public authorities. Sharing the principles and objectives of action. Institutional commitment.	New political leadership at both Ministries (Health and Ecological Transition) Formal and robust structure in order to ensure sustainability of agreements. Political stability and interest Technical committees that support the development of Health Impact Assessments Formalizing intersectoral mechanisms.
Strong points of the PD *(according to participants*)	Inclusion of a member of the Department of Public Expenditure and Reform. Discussion on prices and taxation as tobacco control measures.	Generated a will for continuing working on the topic together and in other scenarios/events. Very good and useful conclusions about what needs to be done and what are the next steps. Representatives from different organizations were present, allowing different approaches and consensus building. Fulfilment of the prior objective of the policy dialog: the approval of the law that restricts the marketing and publicity of inadequate foods and beverages to children in Portugal	Communications, interaction and collaboration between the Ministry of Health, Consumer Affairs and Social Welfare and the Ministry for the Ecological Transition. Implication of key policy makers.
Weak points of the PD *(according to participants*)	More stakeholders required especially patients and public. It could be improved by including service users directly in the policy dialog.	More stakeholders participating would have made discussions richer. Lack of some background and information about the JA CHRODIS PLUS.	Expected more active participation from all audience. Expected more specific actions agreed.

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
