# Peer review of "Shaping Policy on Chronic Diseases through National Policy Dialogs in CHRODIS PLUS"

_ijerph, 2020, doi:10.3390/ijerph17197113_

Round 1
Reviewer 1 Report
POINT 1
Dear Authors,
congratulations on the article. A very interesting topic described in a well-articulated and fluent work.
However, I believe that before the manuscript can be published it is necessary that the bibliography will have to be expanded. Some sentences, within the manuscript, do not have a bibliographic reference and this does not provide the manuscript with the right scientific value it deserves.
Bibliographic entries must be added to the lines: 71,94,146,353,379,475 and 666.
The article could benefit from a few more bibliographical references.
POINT 2
The purpose of the study is not clearly stated, and it to be addressed clearly.
POINT 3
Strength: in general, the study is easy to follow and well summarized, raising the main issues related to the topic. Limitation: although the main issues related to the topic raised by the authors, the report is not well interpreted in terms of the issue raised and the purpose/aim of the review is not clearly illustrated, related to other literature on the topic not well mentioned, in particular, recent studies have not taken into consideration.
Author Response
Dear Reviewer 1, please find attached the cover letter from the co-authors in response to your recommendations.

Reviewer 2 Report
This is a well-written manuscript introducing the policy dialogues among multiple stakeholders on health outcomes in EU member states. The major issue is that it suffered from grammatical mistakes. A period was missing on line 31 and “t” should be “to” in line 48 in abstract. More others could be found in lines 164, 192, 352, 454, 814, 821, 854, and 871. The authors should also be award of mixed usage of past tense, present tense, and future tense, especially when introducing applications in three EU countries in section 3.
This manuscript is well written and almost ready to be published, and thus there is no point-by-point comment. But it is long and has some grammatical errors. My suggestion is a major revision since I need to read their revised manuscript again to make sure these errors are fixed.
Author Response
Dear Reviewer 2, please find attached the co-authors' cover letter in response to your reviewing recommendations.

Reviewer 3 Report
Overall, the paper addresses an interesting topic of the role of policy dialogue in addressing important health-related issues. Unfortunately, although the volume of the manuscript is rather large, it is still not clear what the rationale for the paper is. It looks that surely the Authors have been able to signal the importance of chronic disorders and justify the expenditures of European taxpayer on such initiatives like CHRODIS PLUS project. They also to some extent explored the ways how policy dialogues have been managed in three (arbitrarily selected) national settings (so not in all national settings included in CHRODIS PLUS but only three selected on the arbitrary basis). However, promised “broader lessons” for the use of policy dialogue as an upstream policy-making tool in the future preventative approaches, etc., was not addressed in the paper. It also looks, that although the manuscript contains the description of the policy dialogues in three selected countries, it lacks crucial feedback on feasibility and the use of results of the dialogues. The relationship between policy dialogue and deliberative dialogue was also hardly explained in the paper.
Apart some evident gaps in the fulfilment of the promises declared in the description of aims of the study, in quite many places, the Authors make statements which are not supported by relevant literature (some examples below).
A general methodology of policy dialogue is described, but the way the “broader lessons” were analysed is lacking. So a reader is able to understand policy dialogue and to see how this technique was implemented in three countries. Still searching for the assessment of the broader results of the procedures is futile. The Authors do not provide information about how they would assess the effect of described policy dialogues. Instead, they tend to provide some “evaluation” which apparently is a feedback from the participants of the dialogues in specific countries. But how the feelings of participants translate into “broader lessons” is not clear. The tool for the assessment of the feedback from the participants is not provided.
Some detailed comments:
The aims of the paper given in lines 147-151 are rather unclear. I would suggest less ambiguous formulation of the paper aims if at all, the Authors aim at applying concrete methods to obtain results.
Line 270 – if “the literature suggests…”, please refer to relevant literature.
Line 126-127 – there has been insufficient evaluation – was anybody evaluating earlier this aspect and found scarcity of the “evaluation”? Any references?
In general, the description of the elements of the policy dialogue methodology should be supported with relevant literature (Methods).
The rationale for the selection of examples of the national policy dialogues is rather obscure. In what way do “these three dialogues” represent the wide range of topics covered by the fourteen national policy dialogues? Should we take it for granted? What were the criteria for the selection of three from fourteen national cases? The issue what is representative in this context is another puzzle.
Further, the Authors tend to bombard us with the statement assuring that they have selected the crème de la crème from their cases (“were particularly ambitious” or “followed-through attempts”). But this quality is not evidenced at all. Not mentioning about any concrete criteria for the assessment and how they were measured. Apart from this, one has a feeling that other examples would be carried out without appropriate care. Finally, maybe it would be worth to have feedback on more difficult cases to understand obstacles and risks when performing policy dialogue?
In the Discussion, the Authors claim that they will discuss the main considerations “which were common not only to the three case study countries but across all 14 national policy dialogues”. It rather surprising statement providing that the reader has not got a chance to know more deeply only three specific cases, and not 14 national policy dialogues. It looks like the attempt of discussing results which were not presented in the paper. Such an approach is not understandable and in Reviewer’s opinion, not feasible in a scientific paper.
Subsection 4.1 The Authors tend to suggest that policy dialogue is somehow a tool which results from the adoption of a complex system approach. Still, actually they have not evidenced that the policy dialogue is a part of the complex system analysis. Anyway, nearly the whole section is about complex systems approach, instead of discussing how policy dialogue enables “addressing complexity”. If it is a case (policy dialogue is a complex system methodology tool), this should be evidenced in the Introduction.
The Discussion is generally filled with many generic statements not supported neither by the results originating from the three cases presented in the paper nor from the references which would be cited in the text. Section 4.2 may be a good example of such unjustified statements, e.g. “In some countries, the dialogue focused on examining…”, or “to make their decision, organisers took into consideration…”. No reference to the Results section nor other published sources. It looks like a rather free flow of ideas repeating the main statements from the Introduction and Methods without concrete basis. One could expect in the Discussion, more substantial foundations for provided assertions.
The Conclusion section is organised in a peculiar way now. One would expect those main ideas should be somehow distinguished, e.g. as numbered or bulleted list, from the descriptions.
I would strongly recommend that the Authors adhere to the guidelines on how the references should be described. And providing bare links like in case of reference 24 is surely not acceptable.
Finally, the clarity of the paper would benefit from the more concise style and concrete statements supported with adequate references instead of rather lengthy declarations.
Author Response
Dear Reviewer 3, please find attached the co-authors' cover letter in response to your reviewing recommendations.

Round 2
Reviewer 2 Report
The manuscript was greatly improved by the authors’ revisions. A couple of issues remained to be fixed.
(1) On page 17 the authors said: “This lack of follow-up meetings and processes falling outside of the scope/timeframe of the Joint Action make it challenging to…” In this sentence “make” should be “makes”.
(2) On the same page the authors said: “Understanding how priorities and political landscapes adapt to the context, it is also important that…” This sentence also has grammatical mistake.
Other than those, I don’t think I need to review this manuscript again.
Author Response
Dear Reviewer,
Thank you for taking the time to read our manuscript again. We are pleased that you find this version to be an improvement.
Regarding the issues you have noted, we have corrected both.
(1) On page 17 the authors said: “This lack of follow-up meetings and processes falling outside of the scope/timeframe of the Joint Action make it challenging to…” In this sentence “make” should be “makes”.
In line 655 (now on page 16, due to reformatting the document onto the IJERPH template with line numbers), you will see that the sentence says "makes".
(2) On the same page the authors said: “Understanding how priorities and political landscapes adapt to the context, it is also important that…” This sentence also has grammatical mistake.
In line 664, you will see that we have reworded "to the context" to read "over time" so that the full sentence reads: "Understanding how priorities and political landscapes adapt over time, it is also important that dialogue participants envision ways to gain and maintain political commitments from the necessary policy- and decision-makers". I hope this addresses the grammatical mistake you have indicated?
Other than those, I don’t think I need to review this manuscript again.
Thanks once again for your time and feedback.
Best regards on behalf of all of the authors,
Alison Maassen
Reviewer 3 Report
Thank you for considering my comments. I believe that the paper has been improved considerable and it is now far more readible than earlier.
I understand that Annex will be moved to the Supplemental Materials file. This will require appropriate referring to Supplemental file in the text.
Author Response
Dear Reviewer,
Thank you for taking the time to read our manuscript again. We are pleased that you find this version to be an improvement.
We have now changed all references to Annexes 1, 2, and 3 to Supplementary Table S1, Supplementary Table S2, and Supplementary Questionnaire S3, respectively. We have been in touch with our primary contact at the journal to ensure proper formatting for these references.
Thanks once again for your time and valuable feedback.
Best regards on behalf of all of the authors,
Alison Maassen